# Correlation between Lymphatic Surgery Outcome and Lymphatic Image-Staging or Clinical Severity in Patients with Lymphedema

**DOI:** 10.3390/jcm11174979

**Published:** 2022-08-25

**Authors:** Hirofumi Imai, Shuhei Yoshida, Toshiro Mese, Solji Roh, Asuka Fujita, Ayano Sasaki, Shogo Nagamatsu, Isao Koshima

**Affiliations:** 1International Center for Lymphedema, Hiroshima University Hospital, Hiroshima 734-8551, Japan; 2Plastic and Reconstructive Surgery, Hiroshima University Hospital, Hiroshima 734-8551, Japan

**Keywords:** lymphedema, lymphatic venous anastomosis, lymphoscintigraphy, indocyanine green lymphography

## Abstract

Lymphoscintigraphy and indocyanine green (ICG) lymphography reveal the severity of extremity lymphedema. Lower extremity lymphedema (LEL) index and NECST classification are related to the clinical severity of lymphedema. We aimed to investigate the correlation between lymphatic surgery, lymphatic imaging, and clinical severity in patients with lymphedema. Thirty-five patients with lower-extremity lymphedema who underwent lymphatic venous anastomosis (LVA) were evaluated. Ten of the thirty-five patients underwent multi-surgery (additional vascularized lymphatic transfer and/or liposuction). We investigated the correlation between the LEL index, NECST classification, lymphoscintigraphy staging, ICG lymphography staging, and rate of improvement (RI: [preoperative LEL index − postoperative LEL index]/[preoperative LEL index] × 100). The LEL index in 35 patients after LVA and all procedures decreased significantly compared to that of preoperative (272.4 vs. 256.2 vs. 243.5, *p* < 0.05). RI after LVA and all procedures showed positive correlations with the preoperative LEL index; however, there was no correlation with any other lymphatic image or clinical severity. LVA can reduce lymphedema circumference at any stage. Additional surgery improved the circumference. Hence, LVA as the first line of treatment, and vascularized lymphatic transfer and liposuction as additional procedures, should be considered as the standard treatment for lymphedema.

## 1. Introduction

Extremity lymphedema can be a considerable burden for patients undergoing cancer treatment, as it restricts their daily activities [1]. Lymphoscintigraphy and indocyanine green (ICG) lymphography are the main diagnostic methods for lymphedema, which are useful for evaluating its severity [2]. Both of these tests diagnose lymphedema based on the presence of lymph nodes, lymphatic pathways, and dermal back flow [3,4]. Lymphoscintigraphy and ICG lymphography have now became a gold standard for the diagnosis of lymphedema due to their reliability [2].

The lower extremity lymphedema (LEL) index is calculated by dividing the sum of the squares of the circumference of the five areas of the affected extremity by the body mass index (BMI) [3]. Based on the macroscopic findings during surgery, the collecting lymphatic channels are classified as normal, ectasis, contraction, and sclerosis type (NECST) classification; stages 0–3) [5]. The LEL index and NECST classification clinically reflect the severity of lymphedema and are used worldwide owing to their simplicity.

The treatment of lymphedema consists of conservative treatment, including compression therapy and physical therapy, and surgical treatment, including lymphatic vein anastomosis (LVA), vascularized lymphatic channel transfer (VLT), and liposuction [6]. LVA performed under local anesthesia and VLT and liposuction required general anesthesia; hence, LVA became the first-line of surgical treatment for lymphedema because it is less invasive [7]. VLT is suitable for treating severe lymphedema that is refractory to LVA [8]. Liposuction is indicated for moderate-to-advanced lymphedema, with a significant component of fat hypertrophy. All surgical procedures are effective in reducing the lymphedema severity [9]. However, the correlation between the effectiveness of surgical procedures and lymphatic imaging and clinical severity is unclear.

This study aimed to investigate the correlation between baseline lymphoscintigraphy, ICG staging, or clinical severity and its ability to predict treatment response and prognosis in patients with lower extremity lymphedema who have undergone surgical procedures.

## 2. Materials and Methods

### 2.1. Patients

We performed a retrospective study with 35 patients who underwent lymphoscintigraphy, ICG study, and surgical procedures for lower-extremity lymphedema between April 2018 and December 2021 (Figure 1). The exclusion criteria were: patients who had a lymphoscintigraphy or ICG study staging of 0; those over 80 years of age; those with lymphorrhea or metastasis; and those with follow-up periods of less than 6 months. This study was approved by our Institutional Review Board (approval number: E-1413), and all study participants provided written informed consent.

### 2.2. Technical Aspects of Lymphoscintigraphy

A small amount (0.2 mL, MBq) of technetium-99m-labeled human serum albumin was injected subcutaneously into the first web space of the foot and the lateral malleolus. Using a gamma camera, anterior images of both the lower extremities were obtained immediately after the injection (at 5–10 min), and another set of delayed images were taken at 1 h and 2 h intervals. The images were classified as type I to V, as previously described (Figure 2) [4].

### 2.3. Technical Aspects of Indocyanine Green Study

A small amount (0.1 mL) of ICG (Diagnogreen Injection, Daiichi Pharmaceutical, Tokyo, Japan) was injected subcutaneously into the first web space, lateral malleolus, and the lateral side of the superior edge of the knee [10]. Furthermore, 12 to 18 h after the injection, we observed the ICG lymphography results using a near-infrared camera (Photodynamic Eye; Hamamatsu Photonics, Hamamatsu, Japan) and classified them into types I to V, as previously published (Figure 3) [3].

### 2.4. Clinical Variables

The age, sex, cause of lymphedema (primary or secondary), and preoperative LEL index [11] were obtained through a review of electronic medical records. The LEL index was calculated by dividing the sum of the squares of the circumference in the five areas of the affected extremity by the BMI. The rate of improvement (RI) in lymphedema was calculated by dividing the difference in the LEL index before and after surgery by the preoperative LEL value for each case, as follows: RI = [preoperative LEL index − postoperative LEL index]/(preoperative LEL index) × 100. Additionally, we retrieved the type of surgical procedure performed, the NECST classification [5], and the number of LVAs performed. The LEL index and NECST classification as well as lymphoscintigraphy and ICG lymphography reflect the severity of lymphedema [5,11].

### 2.5. Surgical Procedures

LVA was performed under local anesthesia in all cases along a linear pattern or along the greater saphenous vein course in the area of the dermal back-flow pattern, or without enhancement. The LVA procedures were performed in an end-to-end manner using 11-0 or 12-0 nylon micro sutures under a surgical microscope [12]. Any additional surgeries, such as VLT and liposuction, were performed under general anesthesia. The VLT was elevated from the superficial circumflex iliac artery perforator flap, first metatarsal artery flap, or the lateral thoracic artery perforator flap. The recipient vessel was a perforator of the posterior tibial artery, which was detected preoperatively using ultrasonography; a concomitant or subcutaneous vein was used for drainage [13]. Extensive liposuction was performed while sparing the LVA or VLT sites in order to avoid injury [14]. Circumferential liposuction was performed from the ankle to the hip, and as much of the hypertrophied fat as possible was removed, using the circumferences measured in the healthy limb as reference values. Patients who were not satisfied with the results of the first LVA were advised VLT or liposuction. Before and after surgical treatment of the lymphedema, the patients were advised to wear a compression stocking, which provided approximately 30 mmHg of pressure.

### 2.6. Statistical Analysis

Data are shown as mean and range. Parametric, non-parametric, and categorical variables were compared using Student’s ‘*t*’ test and Wilcoxon’s rank-sum test, respectively. Spearman’s rank correlation was used to evaluate the association between the quantitative indicators. To identify independent risk factors that can predict the reduction of lymphedema severity, all potential risk factors were entered into a multiple regression analysis model. JMP software (version 16.0, SAS Institute Inc., Cary, NC, USA) was used for analyses, and a *p* value < 0.05 was considered statistically significant. The correlation was defined as: weak when r = 0.00–0.19, mild when r = 0.20–0.39, moderate when r = 0.40–0.59, strong when r = 0.60–0.79, and very strong when r = 0.80–1.00.

## 3. Results

### 3.1. Clinical Parameters

The clinical and lymphedema characteristics of the 35 included patients are shown in Table 1. The mean age of the patients was 62.4 years (21–78 years); 22 were women while, 13 were men. Lymphedema was a complication of malignant tumor treatment in 24 patients (68.6%), primary in 6 (17.1%), and caused by cellulitis in 5 (14.3%). The patients with lymphedema caused cellulitis noted the edema after cellulitis. The preoperative LEL index was 272.4 (180.7–384.8). The mean lymphoscintigraphy staging was 3.2 (1–5), and that of ICG lymphography staging was 2.9 (1–5). The mean NECST staging during surgery was 2.1 (0–4). The mean number of anastomoses during first LVA surgery was 5.3 (4–10).

### 3.2. Reproducibility and Correlation of the Clinical Severity Staging and Image Staging

The preoperative LEL index and the NECST classification exhibited moderate positive correlations (r = 0.48, *p* < 0.01) (Figure 4A). Lymphoscintigraphy and ICG lymphography exhibited strong positive correlations (r = 0.62, *p* < 0.001) (Figure 4B). The preoperative LEL index exhibited mild and moderate correlations with lymphoscintigraphy (r = 0.35, *p* = 0.03) (Figure 5A) and ICG lymphography (r = 0.45, *p* < 0.001), respectively (Figure 5B). The NECST classification exhibited a strong positive correlation with ICG lymphography (r = 0.72, *p* < 0.001) (Figure 5C) and a moderate correlation with lymphoscintigraphy (r = 0.47, *p* < 0.01) (Figure 5D).

### 3.3. Correlation between the Results of the First LVA and Lymphatic Images

After the first LVA, the patients had a follow-up for a mean of 11.0 (6–36) months. After the first LVA, the LEL index decreased significantly compared to the preoperative (256.2 (200.8–399.0) vs. 272.4 (180.7–384.8), *p* < 0.01) and the RI was 5.3 (−16.3–28.8). RI after the first LVA and the preoperative LEL index exhibited mild positive correlations (r = 0.42, *p* < 0.05) (Figure 6A). However, RI after the first LVA was not correlated with any of the staging modalities (lymphoscintigraphy, ICG lymphography, or NECST classification) or the number of LVAs (Figure 6B–D) (Table 2).

### 3.4. Results and Comparison of Single LVA with the Multi-Surgery Group

We performed a single LVA in 25 patients (71.4%) and multi-surgery in 10 patients (Table 1). In the multi-surgery group, the patients underwent additional surgery 8.2 (6–12) months after the first LVA. Among the patients who underwent LS, the volume removed was 900 (300–2200) mL. The follow-up period from the last surgery was 24.8 (6–38) months in the multi-surgery group and 15.9 (6–38) months in all patients. In the single LVA and multi-surgery groups: the preoperative LEL index was 271.4 (180.7–384.8) and 274.8 (209.1–358.5) (*p* = 0.97), respectively; lymphoscintigraphy staging was 3.0 (1–5) and 3.8 (1–5) (*p* = 0.26); ICG staging was 2.5 (1–5) and 3.9 (3–5) (*p* < 0.01); and NECST staging was 1.9 (0–4) and 2.6 (0–4) (*p* = 0.06) (Figure 7A–D). In the single LVA group, LEL index decreased after surgery (250.1 vs. 271.4, *p* < 0.01); however, in the multi-surgery group, the LEL index did not significantly change after the first LVA (271.4 vs. 274.8, *p* = 0.93). In comparison, the LEL index after the first LVA was 250.1 and 271.4 (LVA vs. multi-surgery group; *p* = 0.20), and RI after the first LVA was 7.6 (−16.3 to 28.8) and 1.0 (−10.1 to 11.9) (LVA vs. multi-surgery group; *p* = 0.04) (Figure 8A,B). After all the surgical procedures were performed, the final LEL index in the multi-surgery group was significantly decreased to 254.0 (218.1–321.4) when compared to the preoperative and post-first LVA LEL index (*p* = 0.03 for both). The final RI in the multi-surgery group increased to 6.5 (−10.2 to 20.4), which was similar to the results of the single LVA group (Figure 8C).

### 3.5. Overall Results and Correlation with Clinical Parameters

After all the surgical procedures were performed, the final LEL index was 243.5 (200.9–399.0), decreased when compared with the preoperative LEL index (*p* = 0.001). The final RI was 7.3 (−16.3 to 28.8), which was significantly higher than the RI after the first LVA (*p* = 0.03). The final RI and preoperative LEL index exhibited strong positive correlations (r = 0.62, *p* < 0.001) (Figure 9A). However, the final RI was not correlated with any staging of NECST classification, lymphoscintigraphy, or ICG lymphography (Figure 9B–D) (Table 3).

## 4. Discussion

In this study, we demonstrated a correlation between lymphoscintigraphy, ICG lymphography, and the clinical features. First, we demonstrated the reproducibility of image staging and clinical severity of lymphedema. We also analyzed the correlation between lymphatic imaging, such as lymphoscintigraphy and ICG lymphography and clinical severity (the LEL index and NECST classification).

Lymphedema is a chronic, progressive pathological condition that results from impaired lymphatic transportation; therefore, early diagnosis and severity assessment through lymphoscintigraphy or ICG lymphography is imperative [15]. Lymphoscintigraphy or ICG lymphography has become the gold standard for the diagnosis of lymphedema because of its reliability and low invasiveness. Studies have demonstrated significant correlations between the two tests [2,16]. In this study, we demonstrated a strong correlation between lymphoscintigraphy and ICG lymphography that is comparable to previous studies. Pappalardo et al. [17] demonstrated a statistical correlation between image staging and clinical severity of lymphedema, whereas Maclellan et al. [18] described no association between lymphoscintigraphy and the volume of lymphedematous extremity. In the present study, we demonstrated a mild correlation between lymphoscintigraphy staging and preoperative LEL index. Pappalardo’s study included only gynecological cancer-related lymphedema, whereas Maclellan’s study included 79% of patients with primary lymphedema. In the present study, we included both lymphedemas caused by complication of malignant tumor treatment (68.6%) and primary lymphedemas (17.1%). This difference in study population and lymphoscintigraphy findings may be the reason for the different findings in our study. Pappalardo’s study and our study used lymphoscintigraphy staging. In contrast, Maclellan’s study classified the absence or reduction of regional lymph nodes and/or dermal back flow. Surgeons can predict the clinical severity of lymphedema using lymphoscintigraphy staging. Garza et al. [19] demonstrated a mild correlation between ICG lymphography and the volume of lymphedematous extremity (r = 0.33). This finding is comparable to the present study (r = 0.45). NECST classification demonstrated a strong correlation with the International Society of Lymphology staging [5]; however, its correlation with image staging has not been investigated. Despite the strong positive correlation between NECST classification and ICG lymphography, its correlation with lymphoscintigraphy was only moderate. ICG lymphography is a scan of the superficial layer of the collecting lymphatic channels, while lymphoscintigraphy is a scan of the deeper layers of collecting lymphatic channels. We classified the collecting lymphatic channel using the NECST classification based on the findings in the superficial layer; hence, NECST classification had a stronger positive correlation with ICG lymphography than does lymphoscintigraphy.

The reconstruction of the lymphatic drainage route is a viable surgical option for the treatment of lymphedema. LVA has been performed to reroute the lymphatic system to the venous circulation in lymphatic disorders. LVA has been reported to have excellent efficacy in decreasing the extremity circumference and in improving subjective symptoms [20,21]. VLT and liposuction have also been recommended for cases refractory to LVA [8]. In this study, the LEL index improved postoperatively compared to that preoperatively. However, little is known about the factors that influence surgical treatment outcomes. We demonstrated that the LEL index improved regardless of lymphatic image staging or clinical severity. Patients who underwent VLT or liposuction tended to be classified into the more severe lymphedema groups, which included a staging of >3 in lymphoscintigraphy, ICG lymphography staging, and NECST. These results suggest that the greater the severity of lymphedema, the greater the need for additional surgery after the first LVA. Notably, after the additional surgery, the LEL index had improved equally in both groups (single LVA and multi-surgery groups). The lymphedema severity improved, irrespective of the imaging stage and clinical severity. Thus, surgical interventions, with LVA as the first line of treatment and VLT or LS as additional procedures, should be considered as a standard treatment for lymphedema.

This study had some limitations. The study could have an inherent bias as seen in retrospective reviews without randomization. We did not include a control group and the comparison of LVA-VLT and LVA-LS in this study. The inclusion of these groups is required to investigate treatment strategy for lymphedema. An average follow-up of 15.9 months (minimum of six months) was sufficient to assess lymphedema reduction. However, the condition of edema may change six months after surgery and was not captured in the study in some cases. Furthermore, the study was limited by its small cohort size. Thus, large-scale randomized clinical trials are warranted to assess the long-term outcomes associated with either of these techniques.

## 5. Conclusions

We showed a correlation between the results of lymphatic surgery, lymphatic images, and clinical severity of lymphedema. Patients with a staging of >3 in lymphoscintigraphy, ICG lymphography, and NECST tended to require additional surgery (VLT or liposuction) after the first LVA. After all the procedures were performed, the LEL index had sufficiently improved, indicating that surgical interventions for lymphedema, with LVA as the first-line of management and VLT or liposuction as additional procedures, should be used as a standard treatment.

## Figures and Tables

**Figure 1 jcm-11-04979-f001:**
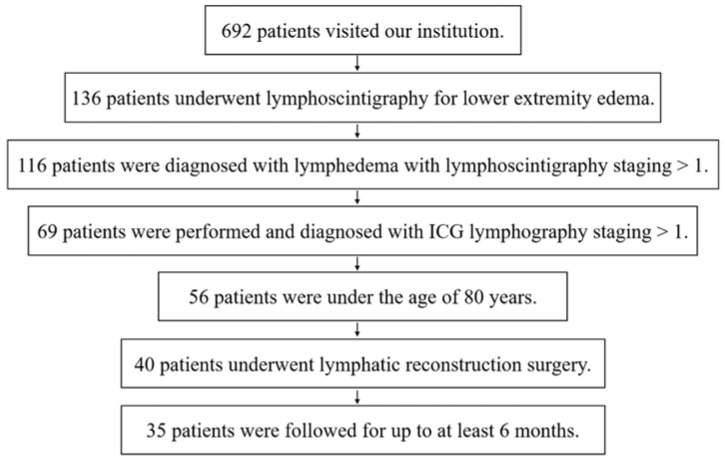
Flow chart of the selection and enrollment process of the patients in this study. ICG, indocyanine green.

**Figure 2 jcm-11-04979-f002:**
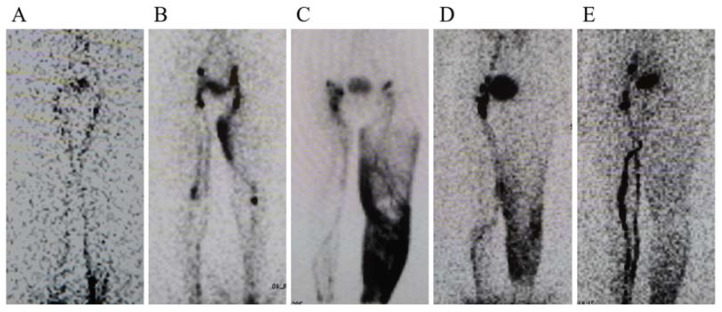
Lymphoscintigraphy images of stages I to V. (**A**) Stage 1: The number of visible inguinal lymph nodes is reduced. (**B**) Stage 2: Few or no inguinal lymph nodes were observed. Dermal backflow was observed in the thigh. (**C**) No inguinal lymph nodes were detected, and dermal backflow was observed in the thigh and/or leg. (**D**) Recognition of dermal backflow and lymph stasis in the lymphatic system of the leg. (**E**) No dermal backflow in the thigh or leg was observed.

**Figure 3 jcm-11-04979-f003:**
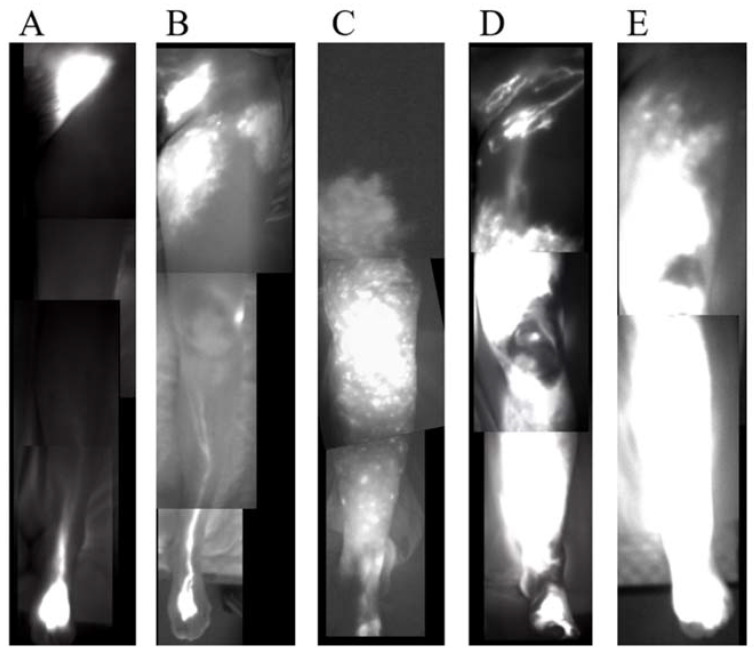
Indocyanine green (ICG) lymphography images of stage I to V. (**A**) Stage 1. Splash pattern around the groin region. (**B**) Stage 2. Stardust pattern extending proximal to the superior border of the patella. (**C**) Stage 3. Stardust pattern extending distal to the superior border of the patella. (**D**) Stage 4. The observed stardust pattern extends to the entire limb. (**E**) Stage 5. The existence of a diffuse pattern with a stardust pattern in the background.

**Figure 4 jcm-11-04979-f004:**
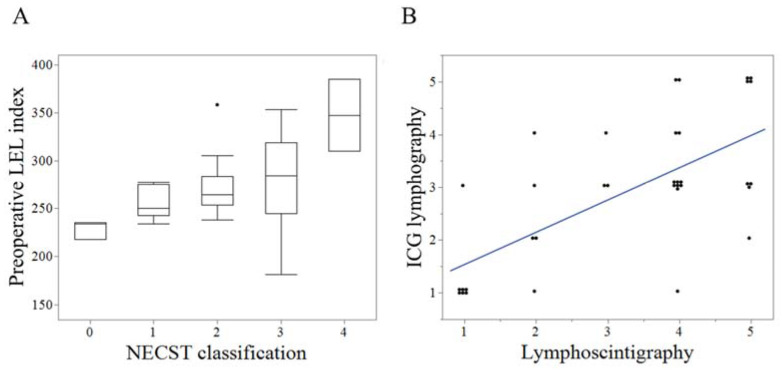
Reproducibility of the clinical severity staging and image staging. (**A**). Box plot of the correlation between preoperative lower extremity lymphedema (LEL) index and indocyanine green (ICG) lymphography stage. (**B**). Scattered plot of the correlation between lymphoscintigraphy and ICG lymphography staging.

**Figure 5 jcm-11-04979-f005:**
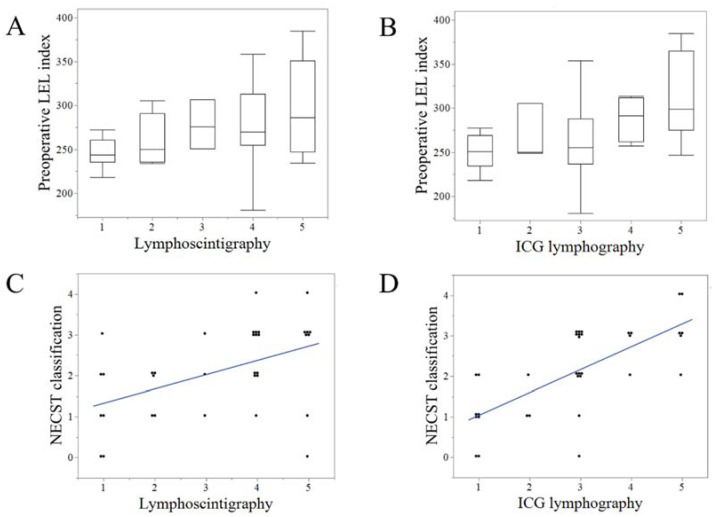
Correlation between the clinical severity staging of lymphedema and lymphoscintigraphy and ICG lymphography. The preoperative LEL index exhibited a mild correlation with lymphoscintigraphy (r = 0.35, *p* = 0.03) (**A**) and moderate correlation with ICG lymphography (r = 0.45, *p* < 0.001). (**B**) The normal, ectasis, contraction, and sclerosis type classification (NECST) exhibited a strong positive correlation with ICG lymphography (r = 0.72, *p* < 0.001) (**C**) and a moderate correlation with lymphoscintigraphy (r = 0.47, *p* < 0.01) (**D**). ICG, indocyanine green; LEL, lower extremity lymphedema.

**Figure 6 jcm-11-04979-f006:**
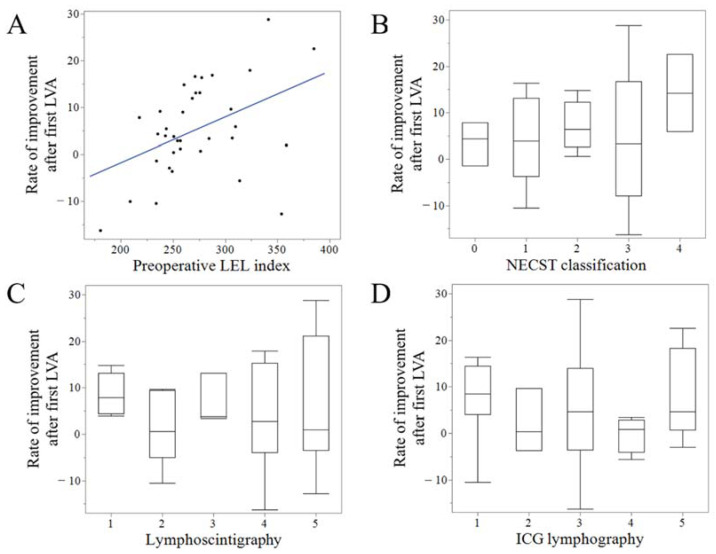
Correlation between the results of the first lymphatic venous anastomosis (LVA) and clinical parameters. Rate of improvement (RI) after the first LVA and preoperative lower extremity lymphedema (LEL) index exhibited moderate positive correlations (r = 0.42, *p* < 0.05) (**A**). However, RI after the first LVA was not correlated with any stage of normal, ectasis, contraction, and sclerosis type classification (NECST) (**B**), lymphoscintigraphy (**C**), or indocyanine green (ICG) lymphography (**D**).

**Figure 7 jcm-11-04979-f007:**
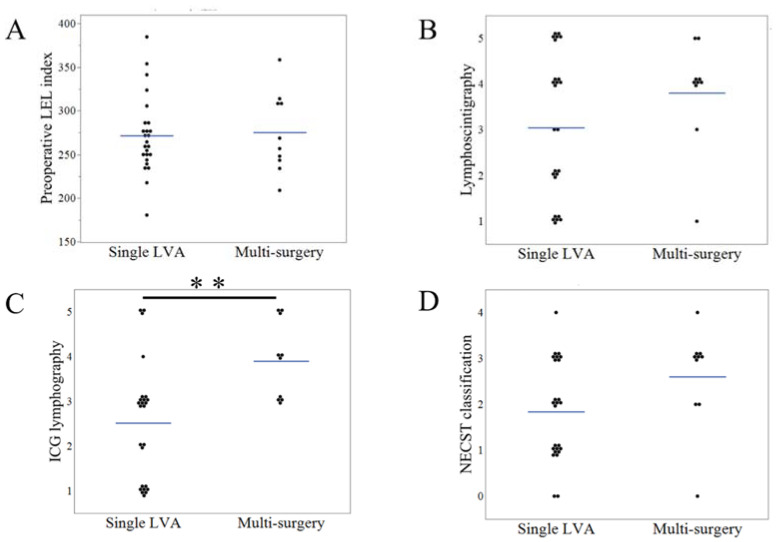
Comparison of clinical parameters in the single lymphatic venous anastomosis and multi-surgery groups. The comparison of preoperative lower extremity lymphedema (LEL) index (**A**), lymphoscintigraphy staging (**B**), ICG lymphography staging (**C**), and the normal, ectasis, contraction, and sclerosis type classification (NECST) staging (**D**), in the single LVA and multi-surgery groups. ** *p* < 0.01.

**Figure 8 jcm-11-04979-f008:**
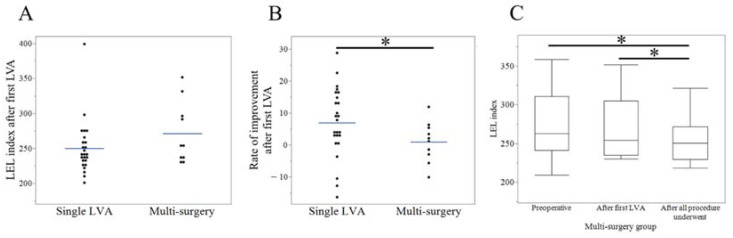
Comparison and results between the single LVA and multi surgery group. Comparison of the lower extremity lymphedema (LEL) index (**A**) and rate of improvement (**B**) after first the LVA between the single LVA and multi-surgery groups. (**C**). The LEL index decreased after all the procedures were performed in the multi-surgery group. * *p* < 0.05.

**Figure 9 jcm-11-04979-f009:**
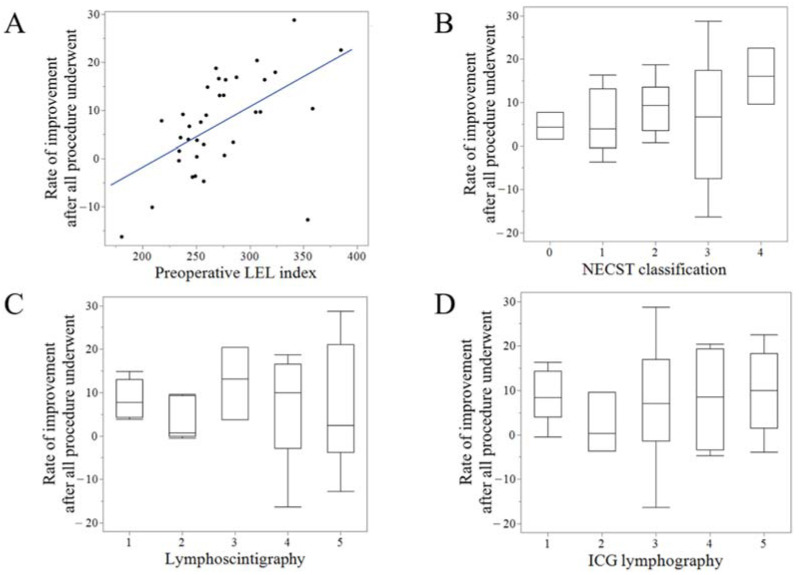
Correlation between the final results of lymphatic surgery and clinical parameters. The rate of improvement (RI) after all the procedures were performed exhibited strong positive correlations with the preoperative LEL index (r = 0.62, *p* < 0.001) (**A**) However, the final RI did not correlate with any staging of the normal, ectasis, contraction, and sclerosis type classification (NECST) (**B**), lymphoscintigraphy (**C**), or ICG lymphography (**D**).

**Table 1 jcm-11-04979-t001:** Clinical characteristics of the patients.

Characteristics	Value
Age, years	62.4 (21–78)
BMI, kg/m^2^	25.7 (16.4–36.4)
Sex, Women/Men	22/13
The cause of lymphedema	
Complication of malignant tumor treatment	24 (68.6%)
Primary	6 (17.1%)
Cellulitis	5 (14.3%)
Duration of edema, m	52.5 (1–360)
≥1 past cellulitis episode(s)	11 (31.4%)
LEL index	272.4 (180.7–384.8)
Lymphoscintigraphy staging	
I	7 (20%)
II	5 (14.3%)
III	3 (8.6%)
IV	12 (34.3%)
V	8 (22.9%)
ICG lymphography staging	
I	8 (22.9%)
II	3 (8.6%)
III	14 (40%)
IV	4 (11.4%)
V	6 (17.1%)
NECST classification	
0	3 (8.6%)
I	8 (22.9%)
II	8 (22.9%)
III	13 (37.1%)
IV	3 (8.6%)
The type of surgery	
LVA	25 (71.4%)
LVA and VLT	5 (14.3%)
LVA and LS	2 (5.7%)
LVA, VLT and LS	3 (8.6%)

BMI, body mass index; LEL, lower extremity lymphedema; ICG, indocyanine green; NECST, normal, ectasis, contraction, and sclerosis type classification; LVA, lymphatic venous anastomosis; VLT, vascularized lymphatic channel transfer; LS, liposuction.

**Table 2 jcm-11-04979-t002:** The multiple regression analysis of factors related to the ratio of improvement after the first LVA.

	Regression Coefficients	*p*-Value	95% Confidence Interval	
			Lower	Upper
Preoperative LEL index	0.13	0.01	0.03	0.23
Episodes of cellulitis	−0.60	0.80	−5.42	4.24
Episodes of radiation	2.53	0.28	−2.15	7.22
Periods of edema	0.01	0.66	−0.03	0.05
Lymphoscintigraphy	−0.92	0.59	−4.40	2.56
ICG lymphography	−0.89	0.68	−5.33	3.56
NECST classification	−0.30	0.90	−5.20	4.60
Number of LVA	−0.97	0.47	−3.72	1.78

LEL, lower extremity lymphedema; ICG, indocyanine green; NECST, normal, ectasis, contraction, and sclerosis type classification; LVA, lymphatic venous anastomosis.

**Table 3 jcm-11-04979-t003:** The multiple regression analysis of the factors related to lymphedema improvement after all surgical interventions.

	Regression Coefficients	*p* Value	95% Confidence Interval	
			Lower	Upper
Preoperative LEL index	0.16	0.001	0.07	0.25
Episodes of cellulitis	0.41	0.86	−4.21	5.02
Episodes of radiation	1.90	0.39	−2.57	6.39
Periods of edema	0.00	0.90	−0.04	0.04
Lymphoscintigraphy	−1.72	0.30	−5.05	1.61
ICG lymphography	−0.17	0.93	−4.42	4.08
NECST classification	−0.73	0.75	−5.42	3.96

LEL, lower extremity lymphedema; ICG, indocyanine green; NECST, normal, ectasis, contraction, and sclerosis type classification.

## Data Availability

The data presented in this study are available on request from the corresponding author.

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
