# Peer review of "Correlation between Lymphatic Surgery Outcome and Lymphatic Image-Staging or Clinical Severity in Patients with Lymphedema"

_jcm, 2022, doi:10.3390/jcm11174979_

Round 1

Reviewer 1 Report

In this study, the authors focus on the surgical strategies for the treatment of patients affected by lymphedema. The aim is to investigate a correlation between the surgical outcome (expressed as rate of improvement), lymphatic imaging and clinical severity.

Overall, the presentation is complete for a scientific paper. The title reflects the content of the paper; the abstract describes the essential information of the work and the introductory section adequately explains the framework and the problems of the research. Tables and graphics are clearly presented.

Below are my specific comments:

Introduction

-Spelling error at line 37 (become instead of became) The authors should clarify the sentence at line 66, as it is not clear whether normal weight or obese patients will be included in the paper.

Materials and methods:

-The nature of the study (observational retrospective) should be soon indicated in the materials and methods section.

- The sentence at line 104-105 seems incomplete (The LEL index and NECST classification reflect the severity of lymphedema, lymphoscintigraphy, and ICG lymphology)

- At line 112 the authors indicate the first metatarsal artery flap as vascularized lymph node flap. (?!)

- At line 119, what does it mean “Patients who were not satisfied with the results of the first LVA”? was a questionnaire administrated to patients in order to evaluate subjective improvement? Was a Threshold of RI used to select patients who need an additional procedure?

Results

- At line 138, cellulitis is indicated as a cause of lymphedema in 14.3% of cases. I don’t believe we could indicate cellulitis as an etiologic factor of lymphedema, but a complication of it. The authors should clarify either in the text and in table 1.

Discussion

In general, one appreciates the authors' willingness to conduct a statistical analysis showing the correlation between the surgical technique used and the initial lymphedema severity or imaging staging as  predictors of outcome. However, some points should be clarified. Regarding the chosen technique, it would seem to be understood that lymph node flaps should only be used in cases of lymphoedema refractory to LVA. This is not the case, since patients with complete occlusion on lymphoscintigraphy and no linear patterns are not candidates for LVA.

Moreover, the research work concludes that patients obtain a benefit after surgical treatment independent of the initial stage. This is nothing new and there are also meta-analyses in the literature that focus more precisely on this point (i.e. PMID: 33721924). I would not go so far as to say " surgical interventions, with LVA as the first-line of treatment and VLT or LS as additional procedures, should be considered as a standard treatment for lymphedema”  (lines 270-272) also because the patients have all undergone different treatments and the work does not prove that the LVA-VLNT sequence is the most correct as there is no control group.

Conversely, what may be more interesting to point out, is the absence of correlation between clinical outcome and initial imaging report. this is a more interesting finding (because it has not been investigated). i would suggest the authors, therefore, revise the discussion in this regard.

Author Response

We thank all the reviewers for their helpful comments. We have carefully considered all the comments and necessary revisions were made in the manuscript accordingly.

We look forward to hearing from you.

Comments: The nature of the study (observational retrospective) should be soon indicated in the materials and methods section.

Response: We thank the reviewer for the careful correction. We have included the study design “retrospective” in the materials and methods section. (Line 59)

Comments: The sentence at line 104-105 seems incomplete (The LEL index and NECST classification reflect the severity of lymphedema, lymphoscintigraphy, and ICG lymphology)

Response: Thank you for the insightful comment. We have revised the sentence as “The LEL index and NECST classification as well as lymphoscintigraphy and ICG lymphography reflect the severity of lymphedema.” (Line 106, 107)

Comments: At line 112 the authors indicate the first metatarsal artery flap as vascularized lymph node flap.

Response: Thank you for the insightful comment. We have used the lymph channel (adiposal) flap (not the lymph nodes) while performing VLT. We have included a reference to support this point.

Koshima, I.; Narushima, M.; Mihara, M.; Yamamoto, T.; Hara, H.; Ohshima, A.; Kikuchi, K.; Todokoro, K.; Seki, Y.; Iida, T.; et al. Lymphadiposal flaps and lymphaticovenular anastomoses for severe leg edema: functional reconstruction for lymph drainage system. J Reconstr Microsurg 2016, 32, 50–55. (Reference 8)

Comments: At line 119, what does it mean “Patients who were not satisfied with the results of the first LVA”? was a questionnaire administrated to patients in order to evaluate subjective improvement? Was a Threshold of RI used to select patients who need an additional procedure?

Response: We thank the reviewer for the careful comment. Patients who underwent an additional surgery were selected based on their satisfaction with the results of the first LVA. We did not use RI as a threshold to determine the necessity of an additional procedure. Patients with severe circumference lymphedema, even though the RI archived was high, were not satisfied with the results of first LVA because they wanted to make their lower extremity thinner. We showed the data that the patients who were not satisfied the result of first LVA were tended to be classified into the more severe lymphedema groups, which included a staging of > 3 in lymphoscintigraphy, ICG lymphography staging, and NECST.

“Patients who underwent VLT or liposuction tended to be classified into the more severe lymphedema groups, which included a staging of > 3 in lymphoscintigraphy, ICG lymphography staging, and NECST. These results suggest that more the severity of lymphedema, the more the need for additional surgery after the first LVA.” (Line 292–296)

Comments: At line 138, cellulitis is indicated as a cause of lymphedema in 14.3% of cases. I don’t believe we could indicate cellulitis as an etiologic factor of lymphedema, but a complication of it. The authors should clarify either in the text and in table 1.

Response: Thank you for the valuable comment. We have added the text “The patients with lymphedema caused by cellulitis noted the edema after the diagnosis of cellulitis.” (Line 141, 142)

We believe that cellulitis can be a cause of lymphedema. The following reference supports this point.

Lin, C.H. Supermicrosurgical lymphovenous anastomosis for the treatment of recurrent cellulitis-associated lymphedema in the lower limb. J Vasc Surg Cases Innov Tech 2021, 7, 790–793.

Patients who had lymphedema after cellulitis were treated with LVA. Lymphoscintigraphy performed before and after LVA showed the improvement of lymph circulation.

Comments about discussion

Response: We thank the reviewer for the valuable comments. We added in limitations that this study does not include the control group and the comparison of LVA-VLT or LVA-LS.

“We did not include a control group and the comparison of LVA-VLT and LVA-LS in this study. Inclusion of these groups is required to investigate treatment strategy for lymphedema.” (Line 302–304)

We have included some sentences in the discussion section regarding the correlation between clinical severity and image staging. We have also included some previous studies and compared their results with that of our’s. (Line 260–276)

Reviewer 2 Report

Several times the authors write ICG lymphology, but it is ICG lymphography,

compression class of the stockings is ok but are they MTM and flat knit?

I never would tell that the first option in lymphedema treatment is a surgical procedure. The follow up of maximum 36 months is not long enough. We see a lot of patients 4-10 years after LVA with increasing lymphedema after 3-4 years, even in combination LVA, lymph node transplantation and liposuction.

Author Response

We thank all the reviewers for their helpful comments. We have carefully considered all the comments and necessary revisions were made in the manuscript accordingly.

We look forward to hearing from you.

Comments: Several times the authors write ICG lymphology, but it is ICG lymphography

Response: We thank the reviewer for the careful correction. We have revised “lymphology” to “lymphography” throughout the manuscript.

Comments: compression class of the stockings is ok but are they MTM and flat knit?

Response: Thank you for the careful comment. The stockings were flat knitted and made by MTM.

Comments: I never would tell that the first option in lymphedema treatment is a surgical procedure. The follow up of maximum 36 months is not long enough. We see a lot of patients 4-10 years after LVA with increasing lymphedema after 3-4 years, even in combination LVA, lymph node transplantation and liposuction.

Response: Thank you for the valuable comment. We have seen cases with deteriorated edema in long-term follow up (4–10 years) after LVA. However, we have also experienced cases in which the edema was decreased and this decrease was maintained for a long-term after LVA. Therefore, we concluded the discussion section as “thus, large-scale randomized clinical trials are warranted to assess the long-term outcomes associated with either of these techniques.” (Line 308, 309)

Reviewer 3 Report

While appreciating the innovative concept of quantifying the results of Lymphedema surgical treatments,
there are some important considerations to discuss, namely:

- The small number of cases

- The very short average period of follow-up considering that, as shown by the literature, the critical issues relating to the outcomes (both of the conservative treatments and of all three indicated surgical treatments) occur even after three, four, five years from the treatment itself.

- There are no guidelines that allow to state that LVA interventions are reserved for simple cases while LTV interventions are reserved for complex ones. The same creator of the VTL technique states that it can be applied in all clinical stages and at a variable distance of time from the onset of the disease.

- What is meant by cellulitis forms (are they primary, secondary?).

- Image detection the day after the ICG injection. Perhaps it is better to specify in hours from injection (with a constant time interval in the various measurements).

- Final consideration: despite the limited number and the average short follow-up, is it possible that improvements have been indifferently detected in all clinical cases?

Author Response

 We thank all the reviewers for their helpful comments. We have carefully considered all the comments and necessary revisions were made in the manuscript accordingly.

We look forward to hearing from you.

Comments: The small number of cases

Response: Thank you for the careful comment. We described in the limitation as you mentioned.

Comments: The very short average period of follow-up considering that, as shown by the literature, the critical issues relating to the outcomes (both of the conservative treatments and of all three indicated surgical treatments) occur even after three, four, five years from the treatment itself.

Response: Thank you for the careful comment. We described in the limitation as you mentioned.

Comments: There are no guidelines that allow to state that LVA interventions are reserved for simple cases while LTV interventions are reserved for complex ones. The same creator of the VTL technique states that it can be applied in all clinical stages and at a variable distance of time from the onset of the disease.

Response: Thank you for your comments. I agree that the guideline has not established so far. We need large scale, long follow study for establishing the guidline.

Comments: What is meant by cellulitis forms (are they primary, secondary?).

Response: Thank you for your comments. We have added the text “The patients with lymphedema caused by cellulitis noted the edema after the diagnosis of cellulitis.” (Line 141, 142)

Comments: Image detection the day after the ICG injection. Perhaps it is better to specify in hours from injection (with a constant time interval in the various measurements).

Response: Thank you for your comments. We use ICG image detection soon after injection for the linear pattern observation which would be masked by following dermal backflow. In this study, We used ICG image detection by only dermal backflow for staging. Hence, we decided image detection the day after the ICG injection was better.

Comments: Final consideration: despite the limited number and the average short follow-up, is it possible that improvements have been indifferently detected in all clinical cases?

Response: Thank you for your comments. As we described in results “The final RI was 7.3 (-16.3 to 28.8)” (Line 236), not all the clinical cases were improved. However, final LEL index was significantly improved when compared with the preoperative LEL index.

Round 2

Reviewer 1 Report

The authors have replied to all my questions, modifying the manuscript according to the suggestions. 

Author Response

We appreciate your comments.

Reviewer 3 Report

I find the new version unchanged from my observations. No specific answers have been given to my observations nor is it emphasized that this is a preliminary study

Author Response

We thank all the reviewers for their helpful comments. We have carefully considered all the comments, and necessary revisions were made in the manuscript accordingly.

We look forward to hearing from you.

Comments: I find the new version unchanged from my observations. No specific answers have been given to my observations nor is it emphasized that this is a preliminary study.

Response: We apologize for the insufficient explanation. Hence, we noted again along with your first comments.

Comments: The small number of cases. The very short average period of follow-up considering that, as shown by the literature, the critical issues relating to the outcomes (both of the conservative treatments and of all three indicated surgical treatments) occur even after three, four, five years from the treatment itself.

Response: In this study, the number of analyzed patients were 35, and the patients were followed up for an average of 15.9 months after surgery. We considered that the size of this study was small; however, it was not too small, and the follow-up period was not very short. For example, long-term outcomes after surgical treatment for lymphedema were reported with 42 patients and 14 months follow-up as the reference.

Alexander T Nguyen , Hiroo Suami , Matthew M Hanasono , Veda A Womack , Franklin C Wong , Edward I Chang. Long-term outcomes of the minimally invasive free vascularized omental lymphatic flap for the treatment of lymphedema. J Surg Oncol. 2017. 115(1). 84-89.

We have noted in line 307 that “the study was limited by its small cohort size.” We noted that the results could be changed after long-term follow-up after surgery in line 306.

Comments: There are no guidelines that allow to state that LVA interventions are reserved for simple cases while LTV interventions are reserved for complex ones. The same creator of the VTL technique states that it can be applied in all clinical stages and at a variable distance of time from the onset of the disease.

Response: We agree that VLT can be applied in all clinical stages and at a variable distance of time from the onset of the disease. We primarily performed LVA, and VLT was performed for the case refractory to LVA, because LVA can be performed with local anesthesia and VLT requires general anesthesia. Regarding invasion, we considered LVA as the first-line surgical treatment for lymphedema. We have added the relevant information to the revised manuscript (lines 48–50).

Comments: What is meant by cellulitis forms (are they primary, secondary?).

Response: Thank you for your comments. We have added the following text: “The patients with lymphedema caused cellulitis noted the edema after cellulitis.” (Line 139-140)

Comments: Image detection the day after the ICG injection. Perhaps it is better to specify in hours from injection (with a constant time interval in the various measurements).

Response: We noted the time interval between ICG injection and image detection, and added the following text to the revised manuscript: “Further, 12-18 hours after the injection, we observed the ICG lymphography results….” (Lines 86–87).

Comments: Final consideration: despite the limited number and the average short follow-up, is it possible that improvements have been indifferently detected in all clinical cases?

Response: Thank you for your comments. As we have described in Results, “The final RI was 7.3 (-16.3 to 28.8)” (lines 229–230), not all the clinical cases improved. However, final LEL index significantly improved when compared with the preoperative LEL index.

Comments: is it emphasized that this is a preliminary study.

Response: Along with comment 2 above, we considered that in this study, the number of patients were not too small and the follow-up period was not very short. Hence, we believe that emphasizing that this is a preliminary study is not required.